# Exploring the acceptability of a community-enhanced intervention to improve decision support partnership between patients with chronic kidney disease and their family caregivers

Shena Gazaway[1,2,3]*, Rachel Wells[1,2], John Haley[4], Orlando M. Gutiérrez[3,5], Tamara Nix-Parker[1], Isaac Martinez[6], Claretha Lyas[3,4], Katina Lang-Lindsey[7], Richard Knight[8], Ruth Crenshaw-Love[9], Allen Pazant[10], J. Nicholas Odom[1,2]

1 School of Nursing, University of Alabama at Birmingham, Birmingham, Alabama, United States of America, 2 Center for Palliative and Supportive Care, University of Alabama at Birmingham, Birmingham, Alabama, United States of America, 3 Nephrology Training and Research Center, University of Alabama at Birmingham, Birmingham, Alabama, United States of America, 4 College of Nursing, Augusta University, Athens, Georgia, United States of America, 5 Heersink School of Medicine, Division of Nephrology, University of Alabama at Birmingham, Birmingham, Alabama, United States of America, 6 Institute for Cancer Outcomes and Survivorship, University of Alabama at Birmingham, Birmingham, Alabama, United States of America, 7 Department of Social Work, Psychology & Counseling, Alabama A & M University, Normal, Alabama, United States of America, 8 American Association of Kidney Patients, Tampa, Florida, United States of America, 9 Montgomery, Alabama, United States of America, 10 Birmingham, Alabama, United States of America

☯ These authors contributed equally to this work.
* gazaways@uab.edu

## Abstract

Patients face numerous health-related decisions once advanced chronic kidney disease (CKD) is diagnosed. Yet, when patients are underprepared to navigate and discuss health-related decisions, they can make choices inconsistent with their expectations for the future. This pilot study, guided by the multiphase optimization strategy and community-engaged research principles, aimed to explore the acceptability of a developed patient component to a decision-support training intervention called ImPart (Improving Decisional Partnership of CKD Dyads). CKD patients and their family caregivers were recruited from an urban, academic medical center. Eligibility criteria for patients included a diagnosis of stage 3 or higher CKD (on chart review), and caregivers participated in interview sessions only. Patients without a caregiver were not eligible. The intervention was lay coach, telephone-delivered, and designed to be administered in 1–2 week intervals for 4 sessions. An interview guide, developed in collaboration with an advisory group, was designed to ascertain participants' experiences with the intervention. Caregiver interviews focused on changes in the patient's decision ability or engagement. Thirteen patients and eleven caregivers were interviewed. The program was viewed as "good" or "beneficial." Three themes capture the intervention's impact– 1) Frequent and deliberate disease-focused communication, 2) Future planning activation, and 3) Coaching relationship. The piloted intervention was successfully

**Data Availability Statement:** Data is shared via the Palliative Care Research Cooperative De-Identified Qualitative Data Repository (QDR-EOLPC). Data will be deposited to the Qualitative Data Repository at the Maxwell School of Citizenship and Public Affairs at Syracuse University - https://data.qdr.syr.edu/privateurl.xhtml?token=50e6171d-1dd6-4146-9a90-be79630c5152.

**Funding:** Research reported in this publication was supported by the Palliative Care Research Cooperative Group, funded by the National Institute of Nursing Research under the National Institutes of Health under Award Number U2CNR014637. The National Institute Of Diabetes And Digestive And Kidney Diseases also supports Dr. Gazaway K23DK134756.

**Competing interests:** The authors have declared that no competing interests exist.

delivered, acceptable to use, and found to promote enhanced disease and future planning communication. By undergoing this work, we ensure that the patient component is feasible to use and meets the needs of participants before implementation in a larger factorial trial.

## Introduction

Patients with advanced chronic kidney disease (CKD) often rely on unpaid family caregivers to help cope and make health-related decisions [1–3]. Family caregivers have been reported to assist their care recipients ≥20 hours/week, performing numerous tasks such as managing medications and symptoms, coordinating care, providing transportation, preparing special diets, and managing household chores [1, 2, 4, 5]. In addition to these tasks, patients face numerous health-related decisions once advanced CKD is diagnosed. These decisions relate to treatment and transplant choices, disease management, when to seek emergency care, employment, insurance and paying for care, obtaining additional help, and desired care at the end of life [2, 4]. When navigating these decisions, caregivers often play various key roles, including seeking out relevant information, facilitating communication, helping patients evaluate the risks and benefits associated with different choices, promoting understanding of available options, eliciting patients' values and preferences, making plans for future decisions, and implementing the chosen course of action [6–11].

Successful health-related decision-making involves understanding complex health information, such as medication regimens, nutritional guidelines, and the underlying pathophysiology of CKD and its progression [12–16]. When dyads are underprepared to navigate and discuss health-related decisions, they can make choices inconsistent with their expectations for the future, values, and preferences [6, 11, 12, 17–19]. Facing these decisions is stressful, and this stress can cause anticipatory grief as patients consider future morbidity and mortality, possibly triggering acute anxiety, sadness, and negative self-image [20]. As decision-making supports, caregivers are faced with helping their care recipients cope with these decision-making experiences, which can be stressful [21, 22].

Systematic reviews and national reports have highlighted several gaps in decision-making. First, interventions focused on optimizing health-related decision-making early in CKD trajectory involving caregivers are lacking [13, 23–26]. Second, studies based on decision-making within a CKD context lack representation from historically excluded populations [27, 28]. Third, existing research has primarily focused on interventions for specific CKD medical decisions such as transplantation, dialysis, location of care, and end-of-life decisions, with limited attention given to the broad range of health-related decisions that arise throughout disease trajectory [16, 23, 26, 29, 30]. Lastly, our own research has demonstrated that patients are making health-related decisions as early as stage 3 and that decision-making is difficult due to lack of information, medical complexity, and poor resource usage [31]. However, social support, such as caregivers, the structure and nature of the medical appointment, and access to resources, were useful in removing barriers to decision-making by enhancing conversation and increasing patients' ability to self-advocate [31]. Given these gaps, we began developing a dyadic decision-support training intervention for patients with advanced CKD and their caregivers. This intervention, ImPart (IMproving the decision PARTnership of CKD dyads), seeks to enhance the skills of social support and communication skills. The tentative intervention would be delivered telephonically by a lay coach navigator for patients with stage 3 or higher CKD. While our group had been refining the caregiver components of ImPart, we shifted focus

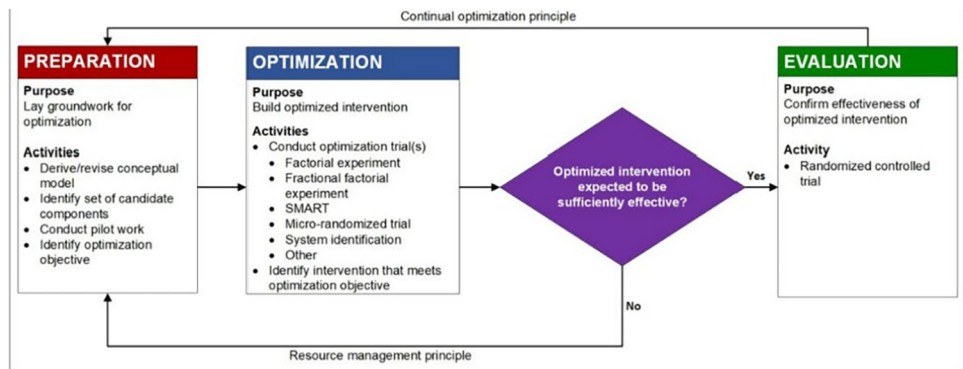

**Fig 1. Flow Chart of the three phases of multiphase optimization strategy (MOST).** Rectangle = action. Diamond = decision. Figure provided with permission from Collins (2018).

based on the qualitative study findings and advice from our advisory group [31]. We began developing a patient component. Hence, this article aims to share the results of a formative evaluation pilot study that explored the acceptability of a patient-focused psychoeducation decision support training component of ImPart. By undergoing this work, we ensure that the patient component is feasible to use and meets the needs of participants before implementation in a larger factorial trial.

## Methods

This study represents the preparation phase of the multiphase optimization strategy (MOST). MOST is a well-validated, comprehensive framework for preparing, optimizing, and evaluating multicomponent interventions [32]. The first phase of MOST, called the preparation phase, is to develop formative work, such as acceptability, feasibility, or pilot testing components of the developing intervention, before moving on to the optimization trial (Fig 1).

This pilot was a 12-month formative evaluation study of ImPart, guided by the MOST, conducted in 2-stages. In stage 1 [July 2022 –October 2022], we utilized collaborative research principles [33] and partnered with a 6-member advisory group of patients, caregivers, and clinicians to develop the patient decision support training component. After approval from the University of Alabama at Birmingham Institutional Review Board (IRB# 300009457), stage 2 [November 2022 –May 2023] consisted of testing the intervention in a single-arm pilot study of two consecutive waves of patient/caregiver dyads. To ensure that diverse experiences were captured, we purposively recruited to ensure that at least 30% of participants identified as Black.

### Stage 1 (July 2022 –November 2022)—community collaboration

The goal of stage 1 was to collaborate with the advisory group and finalize the pilot version of the patient-focused psychoeducation component of ImPart. The advisor board consisted of 2 individuals with CKD, 2 CKD caregivers, and 2 nephrology-focused clinicians, a social worker, and a nephrologist. Advisory board meetings were held using Zoom® every month. The advisory board has a longstanding relationship with the study team, whose members represent patient, caregiving, and clinical experiences in kidney disease from a local, regional, and national perspective. At each meeting, group members were presented with the findings from a previously conducted qualitative study [31] that focused on exploring the decision-making experience of patients with CKD and their caregivers. Discussion and feedback from the

advisory group were used to develop the patient intervention component, including intervention objectives, topical content, and intervention format and delivery.

## Stage II (November 2022—May 2023)—single-arm pilot of the intervention

**Setting and participants.** Participants were recruited from 3 nephrology outpatient clinics of a large, urban academic health science center in the U.S. Southeast. Recruitment began November 7, 2022, and ended April 23, 2023. Ambulatory schedules of partnering nephrologists were screened every 1 to 2 weeks for patients with clinic visits who potentially met the eligibility criteria. Patient inclusion criteria were: 1) age ≥18 years and 2) diagnosed with stage 3 CKD or more severe CKD, including end-stage renal disease and those undergoing dialysis. Exclusion criteria were: 1) uncorrected hearing loss; 2) inability to read or understand English; 3) medical record documentation of a dementia diagnosis; and 4) an untreated psychiatric disorder (e.g., major depressive disorder, schizophrenia).

After gaining nephrologist permission, potential patient participants were mailed a recruitment letter and a copy of the informed consent. The letter described the study and its purpose, objectives, their role in the study should they choose to participate, and compensation for their time. A telephone number was provided if they preferred to opt out of further study contact. Approximately 10–14 days after these letters were sent, a study member called patients and formally invited them to participate. During the recruitment call, patients were asked if they had a caregiver willing to complete an interview after the patient's education sessions had ended. Caregivers participated in data collection only and did not receive intervention sessions. If the patient confirmed the presence of a willing caregiver, the patient verbally consented by S.G or T.NP, and caregiver contact information was collected. If no caregiver was present or would be unwilling to participate, the patient was thanked for their time, and the call was ended.

Caregivers were contacted via phone and provided details of the study, and if interested, were screened for inclusion. Caregiver inclusion criteria were: 1) age ≥18 years; 2) self-endorsement or identification by the patient as "a relative, friend, or partner that has a close relationship with you and who assists you with your medical decisions and who may or may not live in the same residence as you and who is not paid for their help"; 3) caring for a patient with advanced-stage cancer (defined below); 4) have an agreeable patient willing to participate in the study for data collection; and 5) English-speaking and able to complete baseline measures. Exclusion criteria were: self-reported untreated mental illness (i.e., schizophrenia, bipolar disorder, or major depressive disorder), dementia, active suicidal ideation, uncorrected hearing loss, or active substance abuse. If eligible, informed consent was reviewed and verbally collected by S.G. or T.NP, and subsequently, a written copy was mailed.

## The intervention

**Theoretical foundation.** Pearlin's Stress-Health Model of Family Caregiving [34], Rini's Social Support Effectiveness theory [35], and the Ottawa Decision Support Framework [36, 37] provide the conceptual basis of this study (Fig 2).

To keep the model simplified we are not depicting individual and dyadic factors that facilitate stress in caregiving but understand those as social-cultural norms and expectations of caregiver, illness experience, demographic variables, degree of illness, and many others that impact how both individuals in the relationship see the stressor of decision-making. We see where social support effectiveness theory is depicted in the figure as an influencer on individuals' perceived level of support and coping and how decisional needs can be influenced via communication training, a part of the Ottawa framework, and decisional aids. This theoretical

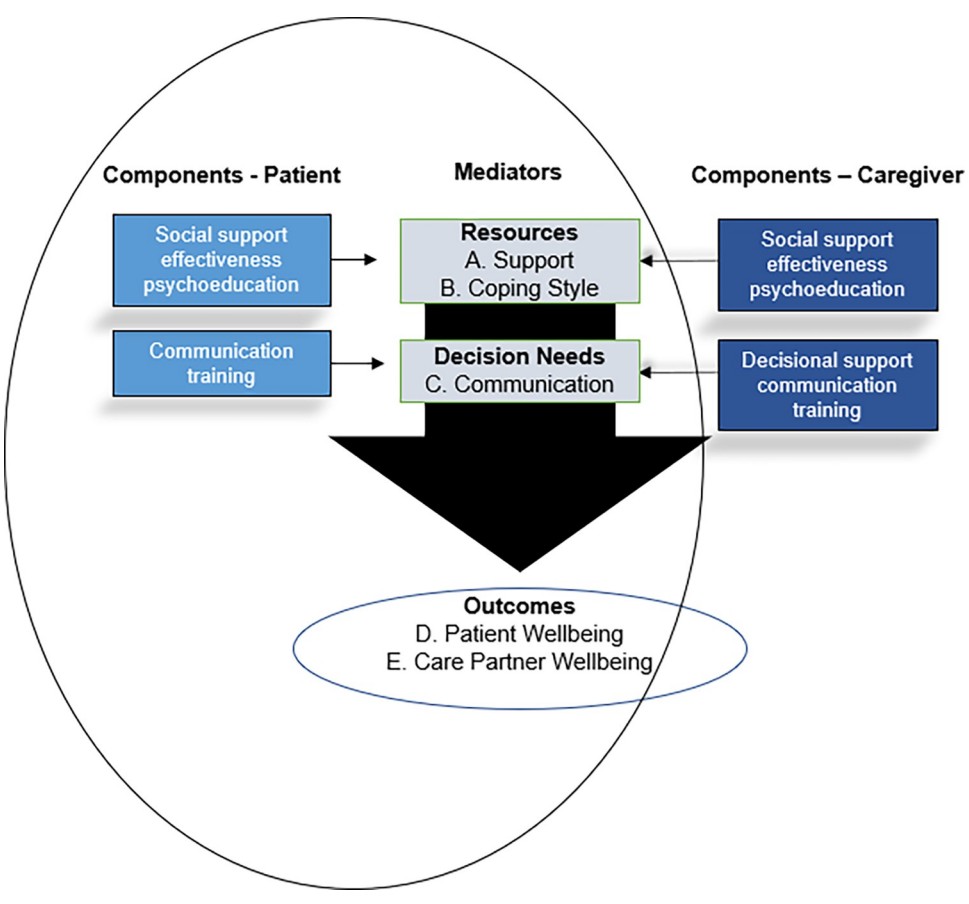

**Fig 2. Conceptual foundation.**

foundation highlights the interconnection of patient and caregiver in the illness experience and decision-making. The goal of this study is to collect data about the dyad's experience and useability of the developed patient training component to determine its potential ability to decrease patients' decisional conflict, enhance perceived decision support, and improve patient (**D**) and caregiver well-being (**E**) (reduce distress, increase quality of life) and (**B**) coping style.

ImPart is conceptualized as a telehealth-delivered psychoeducation program led by a lay coach to enhance patients' skills and enable them to effectively receive and request decision support. Following consent and baseline data collection, the patient was mailed an ImPart toolkit. The toolkit contained core session content, writing space for reflection and notes, and a resource section. The makeup and structure of the toolkit are based on previous work [38, 39]. The toolkit also serves as a medical organization containing a calendar, pen, healthcare professional contact information page, and copies of advanced directives.

The lay coach served in the role of a "health coach." Health coaching uses evidence-based skillful conversations to activate and empower individuals in health behavior change [40, 41]. A systematic review of health coaching in cancer survivors found that the intervention improved quality of life and enhanced patient capacity [42]. For low-income African American women, those randomized in a pilot study to the coaching group experienced a significant reduction in waist circumference than those in the workout group [43]. These studies support the choice to utilize lay coaches as deliveries of the intervention session. To ensure consistency, the lay coach completed 40 hours of structured training, including readings, role play, review,

and observation. The training was designed for individuals with no healthcare background and was adapted from work with lay navigation in cancer care [38, 39].

After confirming toolkit receipt, the lay coach, Male, Hispanic, with prior coaching experience, conducted an introductory call with patient participants. During this call, the coach provided an intervention overview, answered patient participant questions, and scheduled a date and time for session one. Intervention sessions were manualized and designed to be participant-led, with the lay coach as a support to facilitate problem-solving and goal development. After completing the structured sessions, the lay coach conducted monthly follow-up calls for 2 months. There was no provision of direct medical care. Coaches are trained to refer medical or other patient concerns that fall outside their role and scope to the PI, a registered nurse who can debrief. In addition, the PI has access to a clinic-based nephrologist for further consultation as needed.

**Session 1: Recognizing the need for and requesting support.** Participants reviewed and discussed social support effectiveness theory principles, family members' role in serious illness decision-making, and the basic tenets of seeking support and help with healthcare decisions. At the end of this session, participants were introduced and directed to explore two web-based, evidence-based renal replacement decision aids, iChoose [44] and MyTransplantCoach [45].

**Session 2: General communication in advancing illness.** Participants discussed shared decision-making with their nephrologist and strategies to remain active and engaged during their nephrology appointments. Participants could share their thoughts about decision aids and discuss specific questions or conversations they wanted to have with their nephrologist. If desired, these conversations were practiced during the session.

**Session 3. Advanced communication topics.** Participants reviewed and discussed the role of values in decision-making and the role of the family in supporting advanced care planning and advanced directives and in serving as a durable power of attorney. Palliative care as a supportive healthcare option was presented as a case study.

**Monthly follow-up.** Participants received a monthly call for 2 months post-core session completion. During the call, the coach began with a personal interaction, asking about health updates, personal events, or concerns the participant desired to share. The coach then explored any action items from the previous call for which the participant requested follow-up.

## Data collection

After completing each intervention wave, qualitative interviews were conducted with patient participants to ascertain their experiences with the intervention. Interviewers were conducted by 2, Black females over the phone. A semi-structured interview guide Table 1 was developed to explore participants' perceptions of the intervention's acceptability and format. Guide topics included aspects of the component participants liked, found helpful, or disliked; they were also asked if they had changed their behavior related to decision-making or who they spoke with about the program. For caregivers, the interview focused on whether they recognized changes in the patient participant since participating in the program. If they reported engaging with the materials, caregiver participants were also asked how useful they found them.

## Data analysis

Interviews were audio recorded and transcribed verbatim by a professional transcription service (Landmark Associates, Inc). Transcripts were entered into NVivo© qualitative analysis software. Data from transcripts were analyzed using a thematic analysis approach aligned with describing the participant's experience with the intervention [46]. Data analysis occurred as an

**Table 1. Interview guide for patients and caregivers.**

| Type of Question | Patient | Caregiver |
|---|---|---|
| **Opening** | To begin, I would like to hear a little bit about you. Tell me who you are and what you do/did for a living? | Same prompt as patient. |
| | Very briefly and without going into detail, tell me about your chronic kidney disease and the treatment you are currently undergoing. | Very briefly and without going into detail, tell me about your caregiving experience supporting someone with chronic kidney disease and the treatment they are currently undergoing. |
| **Component perception–Now I'd like to hear what you thought of the program.** | How would you describe it to someone who had never heard of it? | Did your family member discuss their involvement in the program–if yes, how would you describe it to someone who had never heard of it? |
| | What did you like most, what made you like those things? What was most helpful? What did you like least, and what made you dislike those things? What was least helpful? | Same prompts as patient |
| **Component impact** | How useful was the session with your coach? Did any of the sessions change something about what you were thinking, feeling, or doing? Please describe. How do you think these sessions impact your ability to manage and make decisions about your chronic kidney disease? | Based on their behaviors around communication–have you noticed a change when you all discuss their kidney disease or when they discuss their disease with their kidney doctor? |
| **Component Structure** | Would you prefer sessions 1 and 2 being combined and focused only on general communication? Or did you like the focus the way it was? | N/A |
| **Coaching—I'd like to hear what you thought about your decision support coach/** | What was your experience like with this person? What things were most helpful about your relationship with him/her? What things could we improve about how you worked with this person? | Did your family member talk about their decision support coach? If yes, I'd like to hear your thoughts about the coaching relationship. |
| **Physical education materials–I'd like to hear what you thought about the educational materials.** | How useful were these materials? What did you like most about it? What would you change or add? | Were you able to see the educational materials? If yes, I'd like to hear what you think about the educational materials. Same prompts as patient. |
| **Involvement** | Are there parts of the program that you have spoken with your caregiver about that you feel would have benefited them as well? | Are there parts of the program you have seen or heard about that you think would have benefitted you if you were involved? |
| **Overall recommendation for change** | What changes you haven't already mentioned should we make to the component moving forward? | Same prompt as the patient |
| **Closing thoughts** | Are there any other closing thoughts that you would like to share concerning your experiences with the ImPart program? | Same prompt as the patient. |

iterative process as interviews were being conducted. Saturation was not a marker of data completeness, as the goal was to interview all patient and caregiver participants. The principal investigator independently listened to the audio recordings and re-read transcripts to immerse herself in the data. Broad categories related to the research questions were developed into codes and used to reduce the raw data. After initial coding, the principal investigator combined similar codes and formed preliminary themes. Preliminary themes and raw text support were shared, discussed, and refined with the study team. Finally, themes were shared with two senior researchers on the project with extensive qualitative expertise to enhance rigor.

## Trustworthiness

The first and fifth authors conducted data collection. The first author, under guidance of the senior author, and the fourth author coded the data. The senior and fourth author are well-established researchers with experience, respectively, in serious illness decision-making, qualitative research methods, kidney disease, and intervention development. Credibility [47] was enhanced through prolonged engagement and the inclusion of the study team, advisory board, and senior researchers in the final conceptualization of themes, supported by rich description

of direct participant quotes. The PI maintained an audit trail to uphold confirmability, including memos, reflections, and bracketing throughout the study. A synopsis of the principal researcher is provided to allow for transparency:

The primary investigator is an early career nurse investigator. Due to her personal and professional experiences in caring for CKD patients and families she has a basic awareness of the diversity in experiences with decision-making in advancing illness. Her motivation for this topic comes from serving as one of two primary caregivers to her grandmother-in-law for 2 years before she died from complications of the disease.

## Results

### Participant characteristics

Twenty-five interviews were conducted with 13 patient participants and 12 caregiver participants. After consenting to be interviewed, one caregiver participant did not return multiple contact attempts from the study team for over a month. Interviews ranged from 15–45 minutes; most lasted approximately 30 minutes. All demographics were self-reported and are shown in Table 2.

Patient participants were mostly White (58.3%), female (76.9%), age greater than 70 (38/5%), and married or widowed (both 30.8%). Patients were mostly retired (53.8%), had a bachelor's degree (30.8%), and had a variety of insurance, including private (38.5%) and private plus Medicare (38.5%). Most patients (58.3%) reported incomes less than $25,000/year. All patients had access to the internet, and a majority (84.6%) reported being comfortable using the internet. Patients reported in the interview experiencing advancing kidney disease; specific CKD stage is not reported here as each participant met eligibility criteria for stage 3 or worse. One patient participant reported currently receiving dialysis.

Caregiver participants were mostly White (58.%), female (91.7%), aged 61–70 (47.7%), and married (58.3%). Caregivers worked full-time (41.7%), had completed some college (41.7%), and had private insurance. In contrast to patient demographics, most caregivers reported incomes higher than $65,000/year and had access to (100%) and comfort (83.3%) using the internet. Most caregivers were the patient's adult child (41.7%), and half lived in the same household.

### Intervention completion feasibility

Sixteen patients verbally consented to begin intervention sessions. Two individuals were withdrawn from the study because they never completed their introduction call with the study coach after their study toolkit was mailed out. One participant completed the introduction call and session one with multiple attempts from the coach but never completed subsequent sessions. We were interested in collecting the perspective of the participant who did not complete sessions to explore their experience with program engagement; those calls and messages were unanswered. Due to the patient's non-completion of the intervention, caregivers were ineligible to participate in the interviews. Our goal was to recruit 10 dyads; we recruited 16 dyads and retained 13 through intervention and interview (8-week process). Thus, we could recruit, engage, and retain 81% of our sample after over-recruiting for a 12-month study.

### Acceptability of impart patient component

Overall, the program was viewed as "*good*" or "*beneficial*" by all patient and caregiver participants. Verbiage supporting these perceptions included "informative" and "enjoyable." The

**Table 2. Participant characteristics.**

| Characteristics | Caregiver | | Patient | | Characteristics | Caregiver | |
|---|---|---|---|---|---|---|---|
| | **n = 12** | | **n = 13** | | | **n = 12** | |
| **Age** | | | | | **Relationship with Patient** | | |
| 21–30 | 0 | (0%) | 1 | (7.7%) | Spouse | 3 | (25%) |
| 31–40 | 1 | (8.3%) | 1 | (7.7%) | Parent | 2 | (16.7%) |
| 41–50 | 2 | (16.7%) | 0 | (0%) | Child | 5 | (41.7%) |
| 51–60 | 2 | (16.7%) | 3 | (23.1%) | Sibling | 2 | (16.7%) |
| 61–70 | 5 | (41.7%) | 3 | (23.1%) | **Lives with Patient** | | |
| 71–80 | 1 | (8.3%) | 2 | (15.4%) | Yes | 6 | (50%) |
| 81–90 | 1 | (8.3%) | 2 | (15.4%) | No | 6 | (50%) |
| 91 or older | 0 | (0%) | 1 | (7.7%) | **Length of Caregiving** | | |
| **Ethnicity** | | | | | Less than 1 year | 5 | (41.7%) |
| Hispanic | 2 | (16.7%) | 0 | (0%) | 1–5 years | 4 | (33.3%) |
| Non-Hispanic | 10 | (83.3%) | 13 | (100%) | 6–10 years | 2 | (16.7%) |
| **Gender** | | | | | Greater than 10 years | 1 | (8.3%) |
| Female | 11 | (91.7%) | 10 | (76.9%) | **How many days/week of Caregiving** | | |
| Male | 1 | (8.3%) | 3 | (23.1%) | 1 day | 3 | (25%) |
| **Religion** | | | | | 3 days | 1 | (8.3%) |
| Yes | 12 | (100%) | 13 | (100%) | 7 days | 1 | (8.3%) |
| No | 0 | (0%) | 0 | (0%) | none | 7 | (58.3%) |
| **Race** | | | | | **How many hours/day of Caregiving** | | |
| Black/African American | 5 | (41.7%) | 4 | (30.8%) | Less than 4 hours | 6 | (50%) |
| White/Caucasian | 7 | (58.3%) | 8 | (61.5%) | 5–8 hours | 1 | (8.3%) |
| Pacific Islander | 0 | (0%) | 1 | (7.7%) | None | 4 | (33.3%) |
| **Marital Status** | | | | | No answer | 1 | (8.3%) |
| Divorced | 2 | (16.7%) | 2 | (15.4%) | | | |
| Married | 7 | (58.3%) | 4 | (30.8%) | | | |
| Single | 3 | (25%) | 3 | (23.1%) | | | |
| Widowed | 0 | (0%) | 4 | (30.8%) | | | |
| **Employment Status** | | | | | | | |
| Fulltime | 5 | (41.7%) | 2 | (15.4%) | | | |
| Part Time | 0 | (0%) | 2 | (15.4%) | | | |
| Unemployed | 1 | (8.3%) | 0 | (0%) | | | |
| Retired | 4 | (33.3%) | 7 | (53.8%) | | | |
| Disabled | 2 | (16.7%) | 2 | (15.4%) | | | |
| **Education** | | | | | | | |
| High School/GED | 3 | (25%) | 3 | (23.1%) | | | |
| Some college | 5 | (41.7%) | 2 | (15.4%) | | | |
| Associate Degree | 0 | (0%) | 2 | (15.4%) | | | |
| Bachelor Degree | 3 | (25%) | 4 | (30.8%) | | | |
| Graduate Degree | 1 | (8.3%) | 2 | (15.4%) | | | |
| **Insurance** | | | | | | | |
| Private | 6 | (50%) | 5 | (38.5%) | | | |
| Medicaid | 0 | (0%) | 1 | (7.7%) | | | |
| Medicare | 2 | (16.7%) | 0 | (0%) | | | |
| Both Medicaid/Medicare | 0 | (0%) | 1 | (7.7%) | | | |
| Veteran's Administration | 0 | (0%) | 1 | (7.7%) | | | |
| Private + Medicare | 3 | (25%) | 5 | (38.5%) | | | |

*(Continued)*

**Table 2.** (Continued)

| Characteristics | Caregiver | | Patient | | Characteristics | Caregiver | | |
|---|---|---|---|---|---|---|---|---|
| | n = 12 | | n = 13 | | | n = 12 | | |
| None | 1 | (8.3%) | 0 | (0%) | | | | |
| **Income** | | | | | | | | |
| Less than 25K | 2 | (16.7%) | 7 | (58.3%) | | | | |
| 25,001–35K | 1 | (8.3%) | 1 | (8.3%) | | | | |
| 35,001–45K | 1 | (8.3%) | 1 | (8.3%) | | | | |
| 45,001–55K | 1 | (8.3%) | 0 | (0%) | | | | |
| 65,001 or higher | 6 | (50%) | 3 | (25%) | | | | |
| No answer | 1 | (8.3%) | 0 | (0%) | | | | |
| **Internet Access** | | | | | | | | |
| Yes | 12 | (100%) | 13 | (100%) | | | | |
| No | 0 | (0%) | 0 | (0%) | | | | |
| **Internet Comfort** | | | | | | | | |
| Yes | 10 | (83.3%) | 11 | (84.6%) | | | | |
| No | 2 | (16.7%) | 2 | (15.4%) | | | | |

impact of the component is captured in three central themes– 1) Frequent and deliberate disease-focused communication., 2) Future planning activation, and 3) Coaching relationship.

**Frequent and deliberate disease-focused communication.** Changes in communication were the highest reported difference between pre to post-program involvement by both patient and caregiver participants. All patients shared that the training prompted them to "*discuss*" and "*communicate more*" with their caregivers. The component was reported to impact patients' ability to "*talk*" and "*share*" with their caregiver, and some reported that they also felt they were a "*better listener*." Direct impact on patient and caregiver communication was captured most eloquently by Patient Participant 15 and her caregiver in separate interviews held by different individuals–

Patient participant 15—*But, you know, dealing with my—my (relationship stated), who doesn't pay attention, which it did help because then when we sat down to talk about the third one, the third session, she had read it, too, so she made a effort to do it right.*

Caregiver participant 15—*Yes, because, uh, I distinctly remember one of the sessions we had that I read about was communication and how we can better communicate. We-we talk a lot anyway, but emotions sometimes get in the way, and this booklet helped us, you know, learn to communicate without lettin' the emotions get in the way. To just, you know, stop and-and remember what we've learned and go about it that way.*

These communication sessions with caregivers involved more than talking about emotions; patient and caregiver participants recalled increased communication about the disease itself; some patient participants had not shared the effects of the disease on their daily lives;

Caregiver Participant 01 remarked–"*She told me a little bit about it. . .and she sent me the video to help me understand what she goes through on a daily basis with her disease. Things that I didn't know and didn't have a clue."* She then went on to share that knowing more helped her empathize because she understood that the patient was not being lazy.

Patient participants also shared plans to communicate with their clinicians differently. These changes included plans to "*ask more questions*," "*seek clarification about disease*," and

"*have more direct conversations.*" Patients reported that the program helped in various ways, including Patient Participant 12 stating–"*I know more to make a better decision,*" and Patient Participant 07's statement that they planned to "*. . .make sure provider knows more about what you are going through.*" However, Patient Participant 01's recall captures the program's impact–"*the program has helped me get confidence back, and also let me know that I'm in charge of my health condition and my, um—the health condition that I achieve.*"

**Future planning activation.**    Patient participants stated that participation in the program impacted how they think about their disease process regarding their future. Many shared that they had not particularly thought about their disease but that participating in the program had them "*thinking about the future*" and "*thinking ahead.*" These future planning conversations occurred with caregivers and others that the patient had decided should be aware of their plans; this is captured by Patient Participant 14 – "*. . .now [I] include my brothers in those conversations and decisions that I've made, uh, concerning me and things going on with my life.*"

Future planning was described in different ways and was not all disease-focused. Those focused on disease spoke about completing the "*advanced directive*" that was provided. Others talked about letting caregivers know their treatment preferences, captured by Patient Participant 04, age 81, when she communicated with her caregivers—"*I don't want a kidney transplant, and I don't want no dialysis.*" Lastly, a few spoke of updating advanced care planning and personal estate documents they had completed years before, ensuring the details aligned with their current desires. Patient participants did not report engaging in these activities independently; all described that these future-focused actions were done in partnership and after conversations with their caregiver(s).

**Coaching relationship.**    All patient and caregiver participants described the coaching relationship as "*helpful,*" "*fantastic,*" and "*good.*" In addition, the coach was described as "*being flexible,*" a "*good listener,*" and provided "*encouragement.*" Caregiver participants reported that their family member was "*comfortable,*" Caregiver Participant 13 recalled their patient participant–"*discussed a lot. . .*" during the coaching sessions. The coach demonstrated actions viewed as compassionate and caring by patient participants, including "*taking notes and following up*" and "*breaking stuff down.*"

**Overall impact and suggestions to improve.**    All patient and caregiver participants reported the program as informative, even those who felt they were not sick enough and those with college degrees (judge, nurse, and master's prepared teacher). Some participants stated that topics were a review for them, but they could still see the benefit for those who do not possess the same knowledge. Patient Participant 04 noted the program *". . .helped me and I can help others.*" Caregiver Participant 14 expressed–"*I'm just really grateful that you guys did it and, you know, um, helped her deal with it, because she just hasn't talked about it for years.*" While the program was reported as acceptable, there were recommendations for improvement. Suggestions for improvement included: increasing resources to support mental health and coping and more caregiver involvement. The reasons provided to support direct caregiver involvement were that knowing more would help them better understand the situation and support the patient. Lastly, there were suggestions around when the program should be implemented; some stated that having something like this as soon as they were diagnosed would be better, while others thought later in the disease (participants who were stable stage 3B reported this).

## Discussion

We conducted a 2-stage formative evaluation study to develop and refine a decision-support training intervention for patients with CKD. The piloted patient ImPart intervention was successfully delivered by phone, acceptable to use, and, per post-intervention qualitative

interviews, found to promote enhanced disease and future planning communication. Our work goes beyond providing additional evidence to support decision-making needing to be shared and patient-centered [7, 48–51] to providing 3 main ways that participation in the patient-focused component influenced the decision-making process. Accounts by patient participants support the idea that focusing on decision-support training instead of providing knowledge intended to impact decision-making can lead to positive changes in the short term, including caregiver involvement, planned efforts to share and engage with clinicians, and desires to plan treatment choices. In addition, caregivers were included in decision-making. These changes are made stronger with reports of improved confidence in ownership of health status and feeling posed to ask questions or directly state one's wishes. Similarly, it is essential to note that the ability of this component to improve and encourage communication is a vital foundation for self-advocacy reported in cancer survivors [52, 53].

The coaching relationship was a powerful additional influence that provided dedicated support to the patient participant in wanting to make impactful changes in how they engage in their disease process. Our formative work supports current literature reporting that lay coaches enhance patient mobilization of existing support networks and increase engagement in healthy behaviors [43, 54]. We recommend that future studies utilizing lay navigators incorporate a measure or means to capture the influence of the coaching relationship on outcomes, and a way to do this is by using MOST and designing a high and low level of the coaching relationship, for example, high-level engagement would be defined as regularly scheduled coaching sessions 1–2 weeks apart, and low-level engagement would be self-guidance of the activity under study [32, 40]. We also suggest researchers explore how interventions impact diverse dyads differently; for instance, do spousal dyads and parent-children dyads need the same decision support tools or something different to meet their complex needs and relational dynamics?

## Future directions

The results presented here support the team's conceptual model (Fig 2), which states that social support effectiveness and communication training impact patients' perceptions that they have more available support from their friends and family. Hence, there is an impact on coping, and there is evidence that it positively influences patient and caregiver well-being. Key to note is the impact on patient's willingness to communicate their needs and wishes to caregivers and clinicians and their spoken decisions to engage in planning for the future. The continuation of this work will revolve around refining the patient component piloted here based on the data highlighted in this study. In addition, we will finalize the design of the caregiver decision-support component specific to kidney care. Future works will pilot the full dyadic intervention (still in the preparation phase of MOST) as a factorial trial to assess these components' feasibility, acceptability, and preliminary efficacy on patient and caregiver outcomes related to decision conflict, coping, and well-being. As a dyadic intervention, patients and caregivers will receive tailored intervention sessions, independently delivered by their lay coach navigator, focused on training them to be better members of a decision-making partnership. As a framework, MOST is effective and efficient as it provides a means to explore the combination of patient and caregiver training at various levels, providing the team with a more comprehensive understanding of how the intervention performs going into the optimization phase (Fig 1).

## Conclusion

Our study demonstrates that engagement with decision-support training can potentially influence patient-initiated discussions about future treatment and advanced care planning with their caregivers. Although this 2-stage formative pilot study using MOST had diverse

representation of patients and caregivers, we acknowledge that Hispanic/Latinx and sexual and gender minority voices are absent. Future works will expand recruitment to increase these populations' representation. This study's findings demonstrate the need for decision-support interventions upstream of advanced illness that train patients and caregivers to be empowered participants in answer-seeking behaviors to enhance their ability to make informed patient-centered decisions.

## Acknowledgments

The team would like to acknowledge the participants who shared their time, and perspectives with us. Without their contribution this work, and others would not be possible.

## Author Contributions

**Conceptualization:** Shena Gazaway, Orlando M. Gutiérrez, J. Nicholas Odom.

**Formal analysis:** Shena Gazaway, Claretha Lyas, Katina Lang-Lindsey, Richard Knight, Ruth Crenshaw-Love, Allen Pazant, J. Nicholas Odom.

**Funding acquisition:** Shena Gazaway.

**Investigation:** Shena Gazaway, Tamara Nix-Parker, Isaac Martinez, J. Nicholas Odom.

**Methodology:** Shena Gazaway, Orlando M. Gutiérrez, Katina Lang-Lindsey, J. Nicholas Odom.

**Project administration:** Tamara Nix-Parker, Isaac Martinez.

**Supervision:** Shena Gazaway, Orlando M. Gutiérrez, J. Nicholas Odom.

**Validation:** Shena Gazaway.

**Visualization:** Shena Gazaway.

**Writing – original draft:** Shena Gazaway, Rachel Wells, John Haley, Orlando M. Gutiérrez, Isaac Martinez, Claretha Lyas, J. Nicholas Odom.

**Writing – review & editing:** Shena Gazaway, Rachel Wells, John Haley, Orlando M. Gutiérrez, Tamara Nix-Parker, Isaac Martinez, Claretha Lyas, Katina Lang-Lindsey, Richard Knight, Ruth Crenshaw-Love, Allen Pazant, J. Nicholas Odom.

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
