## [Decision Letter · Decision Letter 0]

9 Apr 2024

PONE-D-23-34188Exploring the acceptability of a community-enhanced intervention to improve decision support partnership between patients with chronic kidney disease and their family caregiversPLOS ONE

Dear Dr. Gazaway,

Thank you for submitting your manuscript to PLOS ONE. After careful consideration, we feel that it has merit but does not fully meet PLOS ONE’s publication criteria as it currently stands. Therefore, we invite you to submit a revised version of the manuscript that addresses the points raised during the review process.

We look forward to receiving your revised manuscript.

Kind regards,

André Ramalho, PhD

Academic Editor

PLOS ONE

 [Research reported in this publication was supported by the Palliative Care Research Cooperative Group, funded by the National Institute of Nursing Research under the National Institutes of Health under Award Number U2CNR014637.].  

Reviewers' comments:

Reviewer's Responses to Questions

**Comments to the Author**

1. Is the manuscript technically sound, and do the data support the conclusions?

Reviewer #1: Yes

Reviewer #2: Yes

2. Has the statistical analysis been performed appropriately and rigorously? 

Reviewer #1: N/A

Reviewer #2: Yes

3. Have the authors made all data underlying the findings in their manuscript fully available?

Reviewer #1: Yes

Reviewer #2: Yes

4. Is the manuscript presented in an intelligible fashion and written in standard English?

Reviewer #1: Yes

Reviewer #2: Yes

5. Review Comments to the Author

Reviewer #1: Overall:

Thank you for the opportunity to review this manuscript. This was a qualitative evaluation of a community-based intervention pilot that sought to improve decision making support for patients with stage three chronic kidney disease and above. The intervention was a lay-coach, telephone delivered intervention for decision support called ImPART, and this particular manuscript evaluated the acceptability of the patient focused component of the intervention. This feasibility study represented a phase of the larger intervention trial, guided by the multiphase optimizing strategy., a well-validated framework for multi-component interventions.

The manuscript is well written and concise, however there were sections in the manuscript body that switched between present and past tense, e.g. pg 8, 190-193. Please standardize the tenses.

Introduction: The authors clearly describe why they were focusing on patients solely in this study although the larger trial was designed for dyads. However, the introduction as written seems geared towards a caregiver intervention. Expanding the literature review on the benefits and rationale for psychoeducation for patients in structured decision making support/communication coaching, and the use of lay coaches may be helpful to align the introduction to the intervention components.

The use of the theoretical models were appropriate and is as strength and outcomes are clearly defined; it would be helpful to highlight on Figure 2 to map which theories guided the intervention components..

Methods: Study procedures were well described.

Analysis: The analytic plan was appropriate, justified, and supported trustworthiness and transparency.

Results: The diverse representation of the participants interviewed is a strength.

The purpose of the study was acceptability and feasibility but results reported on mostly acceptability; would have expected to see more feasibility discussed regarding recruitment response rate, dropout, attrition, etc.

Future Directions: Results focused on three themes: 1) Frequent and deliberate disease-focused communication., 2) Future planning activation, and 3) Coaching relationship. The statement in page 18, line 380 states that the results support the model that highlights the effectiveness of social support and communication training on support and coping. Social support impacts support is a circular statement, while coping suggests a stress process; would rephrase this statement or perhaps start with line 381 instead.

There was a high representation of children caregivers, would the intervention be different with partners? Was the intervention applicable across various caregiver relationships?

There is a bit of overlap with future directions and the conclusions paragraph. It also will be helpful to understand how this patient focused component fits with the caregiver component. Perhaps also discussing why MOST is useful here, for example if it is effective just focusing on one member of the dyad, future interventions can be more targeted while maintaining effectiveness.

Reviewer #2: The pilot study addresses a fundamental area of health that, in most cases, is omitted - the training of the sick individual to include the management of the therapeutic regimen in their self-care, making decisions that have an impact on their health.

As an improvement to the method, it is suggested that in future studies, in order to increase the reliability of the qualitative data analysis, clients and their family caregivers could have been consulted regarding the interpretation given to their words.

Greater participation by family caregivers is also suggested, as mentioned in the study conclusions.

6. PLOS authors have the option to publish the peer review history of their article (what does this mean?). If published, this will include your full peer review and any attached files.

Reviewer #1: **Yes: **Djin Lyn Tay

Reviewer #2: No

---

## [Author Response · Author response to Decision Letter 0]

24 May 2024

Journal Requirements 

Formatting and author affiliations requirements The manuscript and author affiliations have has been reviewed and comply with PLOS ONE formatting requirements. 

Financial disclosures The role of the funders has been updated to state the approved language from the funder - The research reported in this publication was supported by the Palliative Care Research Cooperative Group funded by the National Institute of Nursing Research under the National Institutes of Health under Award Number U2CNR014637. The content is solely the responsibility of the authors and does not necessarily represent the official views of the National Institutes of Health: this is located in the cover letter as requested but must be placed in the final published document per funder's requirement. The additional sentence The funders had no role in study design, data collection, and analysis, the decision to publish, or manuscript preparation has also been added to clarify since it is not explicit in the sentence provided by the funders. 

Ethics statement The ethics statement is in the methods section. The second statement at the end of the manuscript has been removed. 

Data availability The data is available through the Palliative Care Research Cooperative QDR Collective here - https://data.qdr.syr.edu/privateurl.xhtml?token=50e6171d-1dd6-4146-9a90-be79630c5152

References References have been reviewed. All are present; there are not retracted references. 

Reviewer 1 

The manuscript is well written and concise, however there were sections in the manuscript body that switched between present and past tense, e.g. pg 8, 190-193. Please standardize the tenses. The entire manuscript has been reviewed for tense. Tense has been updated to the past, where needed throughout (everything before discussion). Thank you. 

The authors clearly describe why they were focusing on patients solely in this study although the larger trial was designed for dyads. However, the introduction as written seems geared towards a caregiver intervention. Expanding the literature review on the benefits and rationale for psychoeducation for patients in structured decision making support/communication coaching, and the use of lay coaches may be helpful to align the introduction to the intervention components. We agree, and the foundational study that urged our development of this patient component is published; we have been able to add this data to the background section. 

Lay coach rationale is shared in The Intervention section, and we would like to keep it that way to keep the why for intervention components separate from the why for doing the study. To address the comment, we added additional support articles for why we are using lay coach navigation to the 2 articles cited, including a systematic review on health coaching and its benefits (page. 8). 

The use of the theoretical models were appropriate and is as strength and outcomes are clearly defined; it would be helpful to highlight on Figure 2 to map which theories guided the intervention components. Thank you. We added two sentences describing the interconnection of the theories and how they fit in regard to the models on page 7. 

The purpose of the study was acceptability and feasibility but results reported on mostly acceptability; would have expected to see more feasibility discussed regarding recruitment response rate, dropout, attrition, etc. Our shared aims are: … to share the results of a formative evaluation pilot study that explored the acceptability of a patient-focused psychoeducation decision support training component of ImPart (31). By undergoing this work, we ensure that the patient component is feasible to use and meets the needs of participants before implementation in a larger factorial trial (pg. 4). Thus, we did not explore protocol feasibility as described in the reviewer's comment. We focused on the feasibility of use. However, we can address details about attrition and dropout once a participant consented to the study. These details were added in the results and Study Process Feasibility sections. 

Results focused on three themes: 1) Frequent and deliberate disease-focused communication., 2) Future planning activation, and 3) Coaching relationship. The statement in page 18, line 380 states that the results support the model that highlights the effectiveness of social support and communication training on support and coping. Social support impacts support is a circular statement, while coping suggests a stress process; would rephrase this statement or perhaps start with line 381 instead Thank you for this note. I understand the confusion. We have rewritten the sentence to emphasize that undergoing social support and community training appears to impact the patient's perception that they have support available from friends and family members, impacting one's ability to cope better. This restructured sentence is on page 18. 

There was a high representation of children caregivers, would the intervention be different with partners? Was the intervention applicable across various caregiver relationships? Thank you for this note. We think this is a great future research area as we do not know how different types of dyads will respond to training. We know there was an impact with spousal partners and other partners. However, this pilot will be limited to finding those associations. The reflection of this question has been added to future research questions/implications on page 18.

There is a bit of overlap with future directions and the conclusions paragraph. It also will be helpful to understand how this patient focused component fits with the caregiver component. Perhaps also discussing why MOST is useful here, for example if it is effective just focusing on one member of the dyad, future interventions can be more targeted while maintaining effectiveness. We agree with this statement. On page 19, at the end of the future directions section, we have added a brief description of the dyadic intervention (as we are starting that work now) and how MOST will assist in setting this work up for success. 

Reviewer 2 

As an improvement to the method, it is suggested that in future studies, in order to increase the reliability of the qualitative data analysis, clients and their family caregivers could have been consulted regarding the interpretation given to their words. Thank you. We will ensure to address this in future studies. Since this is not written as a critique and we can not address it now, we will not add a tacked item to the submission to address this review. 

Greater participation by family caregivers is also suggested, as mentioned in the study conclusions. Thank you, since this is reflected, we will not address this in the resubmission.

---

## [Editor Report · Decision Letter 1]

28 May 2024

Exploring the acceptability of a community-enhanced intervention to improve decision support partnership between patients with chronic kidney disease and their family caregivers

PONE-D-23-34188R1

Dear Dr. Gazaway,

We’re pleased to inform you that your manuscript has been judged scientifically suitable for publication and will be formally accepted for publication once it meets all outstanding technical requirements.

Kind regards,

André Luis C Ramalho, PhD

Academic Editor

PLOS ONE

---

## [Editor Report · Acceptance letter]

31 May 2024

PONE-D-23-34188R1 

PLOS ONE

Dear Dr. Gazaway, 

I'm pleased to inform you that your manuscript has been deemed suitable for publication in PLOS ONE. Congratulations! Your manuscript is now being handed over to our production team.

Kind regards, 

on behalf of

Prof. Dr. André Luis C Ramalho 

Academic Editor

PLOS ONE